# Eicosanoid Content in Fetal Calf Serum Accounts for Reproducibility Challenges in Cell Culture

**DOI:** 10.3390/biom11010113

**Published:** 2021-01-15

**Authors:** Laura Niederstaetter, Benjamin Neuditschko, Julia Brunmair, Lukas Janker, Andrea Bileck, Giorgia Del Favero, Christopher Gerner

**Affiliations:** 1Department of Analytical Chemistry, Faculty of Chemistry, University of Vienna, 1090 Vienna, Austria; laura.niederstaetter@univie.ac.at (L.N.); benjamin.neuditschko@univie.ac.at (B.N.); julia.brunmair@univie.ac.at (J.B.); lukas.janker@univie.ac.at (L.J.); andrea.bileck@univie.ac.at (A.B.); 2Department of Inorganic Chemistry, Faculty of Chemistry, University of Vienna, 1090 Vienna, Austria; 3Joint Metabolome Facility, Faculty of Chemistry, University of Vienna, 1090 Vienna, Austria; 4Department of Food Chemistry and Toxicology, Faculty of Chemistry, University of Vienna, 1090 Vienna, Austria; giorgia.del.favero@univie.ac.at; 5Core Facility Multimodal Imaging, Faculty of Chemistry, University of Vienna, 1090 Vienna, Austria

**Keywords:** batch variations, eicosanoids, fetal calf serum, mass spectrometry, peroxisomes, proteomics

## Abstract

Reproducibility issues regarding in vitro cell culture experiments are related to genetic fluctuations and batch-wise variations of biological materials such as fetal calf serum (FCS). Genome sequencing may control the former, while the latter may remain unrecognized. Using a U937 macrophage model for cell differentiation and inflammation, we investigated whether the formation of effector molecules was dependent on the FCS batch used for cultivation. High resolution mass spectrometry (HRMS) was used to identify FCS constituents and to explore their effects on cultured cells evaluating secreted cytokines, eicosanoids, and other inflammatory mediators. Remarkably, the FCS eicosanoid composition showed more batch-dependent variations than the protein composition. Efficient uptake of fatty acids from the medium by U937 macrophages and inflammation-induced release thereof was evidenced using C13-labelled arachidonic acid, highlighting rapid lipid metabolism. For functional testing, FCS batch-dependent nanomolar concentration differences of two selected eicosanoids, 5-HETE and 15-HETE, were balanced out by spiking. Culturing U937 cells at these defined conditions indeed resulted in significant proteome alterations indicating HETE-induced PPARγ activation, independently corroborated by HETE-induced formation of peroxisomes observed by high-resolution microscopy. In conclusion, the present data demonstrate that FCS-contained eicosanoids, subject to substantial batch-wise variation, may modulate cellular effector functions in cell culture experiments.

## 1. Introduction

Problems with the inter-laboratory reproducibility of results obtained with in vitro cell culture models are increasingly being recognized [1,2]. The need to reduce the use of animal models for research purposes relies also on the use of accurate in vitro test models [3,4]. Important decisions such as the choice of drug candidates to be evaluated in clinical studies may be based on such experiments [5]. Thus, the identification of influencing factors potentially modulating such in vitro data is mandatory. Biological materials and reference materials have been recognized as the main contributors for irreproducibility, resulting in a current focus on the investigation of genetic heterogeneity and genetic instability of cell culture models [6,7]. Here we present Fetal Bovine Serum (FBS; also fetal calf serum, FCS) as another relevant contributor to reproducibility issues. FCS is commonly used as cell culture supplement sustaining the growth and duplication of mammalian cells in vitro. Since its introduction in the 1950s its use has been established world-wide, irrespective of evident limitations regarding scientific as well as ethical points of view [8]. Fetal serum is basically a by-product of meat production collected from the still beating hearts of living fetuses. While efforts are made to reduce the use of FCS, they have shown rather limited successes [9]. 

As for other supplements of natural origin, the main variability source associated to FBS can be traced back to largely uncharacterized bioactive components. Due to low concentrations or lack of experimental standard measurements, they may remain poorly controlled, but may still influence the outcome of cell-based experiments. While some effort has been spent to define the composition of FCS, the main bioactive constituents subjected to meaningful variation are hardly known [8]. Batch-dependent variations have been described to affect biological outcomes but such considerations remain limited to rather specialized topics such as hormone regulation [10]. Chemically defined media (CDM) represent a general and consequent solution for these problems, but have only been established and available for a limited number of cell model systems [9]. 

The focus of the present study was to investigate whether it was possible to identify bioactive compounds in FCS accounting for relevant batch-specific effects and correlate this information with proven and biochemically evident readouts on cell functions. Variations of amino acid and metabolite composition of FCS may be considered as less likely as they should be subjected to the homoeostatic control of the organism and, further limited by the dilution of FCS in the accurately produced cell culture media also containing these molecules (typically 5–10% FCS is used). Thus proteins as well as eicosanoids and other polyunsaturated fatty acids (PUFAs) are profiled as the most relevant bioactive candidate molecules to account for inter-batch variation. Indeed, they are responsible for regulating biological processes associated with inflammation [11] and inflammation-associated pathomechanisms [12,13,14]. The monocyte cell line U937, a well-established cell model for macrophages [15,16,17], was chosen for these investigations. Overall, the data demonstrated significant effects of FCS-contained eicosanoids with batch-dependent variations on relevant cell functions, proving that bioactive lipid content in serum contributes to reproducibility issues in cell culture experiments.

## 2. Results

Formation of bioactive pro-inflammatory mediators by macrophages may be influenced by FCS batch effects.

In order to systematically investigate cell culture reproducibility issues resulting from FCS batch effects, a proteome profiling experiment using a U937 macrophage differentiation and activation model was performed. A single batch of U937 cells was seeded into 24 identical aliquots, forming four groups subsequently sub-cultured with four different FCS batches (Table 1, see Materials and Methods). All cells were differentiated using phorbol 12-myristate 13-acetate (PMA) to induce macrophage formation as verified by FACS (fluorescence activated cell sorting) analysis (Appendix A), while three aliquots of each group (FCS batch) were subsequently treated with lipopolysaccharides (LPS) to induce inflammatory stimulation, the other three per group serving as untreated controls. The formation of inflammatory mediators was investigated by comparative secretome analysis resulting in the identification of 488 proteins (Appendix A) and 54 eicosanoids and fatty acid precursor molecules (Appendix A). Whereas most molecules such as the chemokine CCL3 and CXCL5 showed rather little variation between the groups, reproducibility issues of differentiated U937 macrophages were evidenced by FCS batch-dependent significant (FDR (false discovery rate) < 0.05) differences in the formation of the chemokine CCL5, the cell growth regulator IGFBP2, and the cell migration and fibrinolysis regulator SERPINE1 (PAI1) and MMP1 (Figure 1A). In line, the amount of bioactive eicosanoids comprising the hydroxyeicosatetraenoic acids 11-, 12-, and 15-HETE, hydroxydocosahexaenoic acid 17-HDoHE, the prostaglandin PGJ2 and others were found to differ significantly (FDR < 0.05) depending on the FCS batch used for cell culture (Figure 1B). 

Inflammatory stimulation with LPS induced the secretion of a total of 67 proteins (FDR < 0.05, Appendix A), including tumor necrosis factors TNF and TNFSF15, chemokines such as CCL3, CXCL5, CXCL10, metalloproteinases including MMP1 and MMP10 and other promoters and mediators of inflammation (Figure 1A and Appendix A). Only 22 of those 67 proteins were found uniformly regulated independent of the FCS batch (Figure 1C), whereas other bioactive molecules such as IGFBP2 and TNFSF15 again showed FCS batch-dependent expression patterns (Figure 1C). Similarly, LPS treatment induced the formation of lipid mediators of inflammation such as 15-HETE, PGE2, PGJ2 and others (Figure 1B, Appendix A). The formation of eicosanoids varied rather strongly dependent on the FCS batch used, four out of seven LPS-induced eicosanoids showing significant batch-dependent alterations (FDR < 0.05, Figure 1D). 

### 2.1. The Eicosanoid Content of FCS Varies in a Batch-Dependent Fashion

The induction of inflammatory activities of cells may be subject to modulation by a delicate balance of pro- and anti-inflammatory molecules. Thus we investigated whether the above-described batch effects may be caused by differences of the protein and eicosanoid content of the FCS batches used for the cell culture experiments. Remarkably, the protein profile comprising 289 identified proteins (FDR < 0.01) of the four different FCS batches was rather consistent (Figure 2A, Appendix A). Several significant abundance differences between batches were observed (Appendix A), and a principle component analysis showed fairly good clustering of the FCS samples according to batches (Figure 2B). The analysis of FCS eicosanoid contents revealed even stronger batch-dependent differences, comprising mainly COX and LOX-products (Figure 2C, Appendix A). Here, an unbiased PCA clustered the FCS batches with clear distances between batch clusters (Figure 2D), and demonstrated that batch dependent differences of eicosanoid content exceeded the differences of protein content.

### 2.2. Cell Culture Subjects Fatty Acids to a High Turnover 

Growing cells require medium supply of fatty acids and fetal calf serum is rich in polyunsaturated fatty acids. In order to mediate biological effects as assumed for the FCS-contained fatty acids described above (Figure 2B), cells are supposed to take up fatty acids from the medium. In order to estimate to what extent cultured U937 cells may be able to take up fatty acids from the medium and release fatty acids back into the medium upon stimulation, we investigated the intracellular to extracellular exchange dynamics of the eicosanoid precursor molecule arachidonic acid (AA). For this purpose, cultured U937 cells were spiked with stable isotope labelled AA at a concentration double that of the endogenous AA (1.6 µM). Stable isotope labelled AA can be clearly distinguished from endogenous AA by mass spectrometry. As demonstrated in Figure 3, upon differentiation to macrophages, U937 cells apparently picked up more than 99% of the labelled and spiked AA within 72 h and less than 1% C13-AA remained detectable in the medium after that period. As expected, subsequent LPS treatment triggered a significant increase of the amount of C13-AA in the supernatant (Figure 3). This demonstrated that phospholipase activity was capable of releasing previously incorporated AA back into the medium. When supplementation with C13-AA was performed after PMA differentiation but before LPS treatment, the outcome was similar. Around 95% of the available AA was incorporated, but still a significant release of C13-AA was observed upon LPS treatment, clearly indicating a high turnover rate of AA. Concomitant measurement of endogenous C12-AA confirmed that AA was consumed substantially during cell culture and released back in the medium again in a smaller proportion upon LPS treatment.

### 2.3. Supplementation of 5-HETE and 15-HETE in the Nanomolar Concentration Range Induces the Formation of Peroxisomes in U937 Macrophages

In order to demonstrate that the detected differences in eicosanoid content of FCS could originate from the observed batch effects, we performed an additional proteome profiling experiment with U937 cells at conditions only differing with regard to two selected eicosanoids, 5-HETE and 15-HETE. To this aim, we supplemented the FCS batch found to have the lowest levels of these two eicosanoids, here designated FCS-B (containing 5 nM 5-HETE and 12 nM 15-HETE), with the pure chemicals to levels close to those observed in case of FCS batch A (FCS-A, containing 42 nM 5-HETE and 49 nM 15-HETE, Figure 4A). Thus, U937 cells were grown and differentiated as before, using either FCS-A, supplemented FCS-A, or FCS-B, and subsequently subjected to proteome profiling of cytoplasmic proteins (Figure 4B). Indeed, spiking in of the two HETEs was associated with distinct proteome alterations (Figure 4B) including down-regulation of PKM and up-regulation of PEX 16, a peroxisomal membrane biogenesis protein [18]. To independently verify with a complementary method that this was a relevant observation, peroxisome formation was analyzed using immunofluorescence with an anti-PMP70 antibody. Nuclei and mitochondria were additionally stained to demonstrate uniform appearance of these organelles serving as background control. Indeed, treatment of U937 cells with increasing concentration of HETEs induced the formation of peroxisomes in an apparently concentration dependent manner (Figure 4C). 

## 3. Discussion

The present data demonstrates that variations in the eicosanoid content of FCS may account for substantial batch effects with regard to functional readouts of a cell culture model reporting inflammatory mediators. This finding may be of great relevance for a large number of laboratories working with cell culture and using FCS, as FCS-contained eicosanoids have hardly been considered to have major implications for cell culture experiments and have thus, to the best of our knowledge, not yet been subjected to rigorous control. There are reasons, why relevant effects of eicosanoids contained in FCS were hardly expected. First, these molecules are generally considered to be short-lived and to act mainly in situ [19]. Second, eicosanoids were detected in FCS in the lower nanomolar concentration range, this is much less than the concentration range applied for functional assays in vitro, which is typically around 1 µM [20,21]. Furthermore, fatty acids including eicosanoids contained in serum are bound to albumin and only about 0.1% is actually free from associated molecules [22,23,24]. This free pool has a high turnover rate of about 2 min accounting for the redistribution of albumin-bound fatty acids in vivo to distant organs such as muscles or the liver. 

When investigating FCS batch effects, we initially expected proteins to represent the most plausible candidates contributing to reproducibility issues. Proteins as well as metabolites are strictly regulated in vivo to ensure homeostasis and consequently stable viability of the organism. While proteins may be rather stable in biological environments, metabolites such as fatty acids are much more vulnerable to chemical reactions such as oxidation, which may occur also during processing of FCS and are hard to control. Thus it was somewhat unexpected to see that FCS eicosanoid profiles were stable and clustered the FCS samples according to batches (Figure 2). This finding, supported by older and current literature reporting remarkable biological effects of eicosanoids [25,26,27], motivated us to focus on this class of molecules. Functional analyses were based on spiking experiments with the U937 cells. As a first step, the efficient and fast uptake of albumin bound arachidonic acid (AA) was verified in the present cell model system using stable isotope labelled AA. The subsequent release of labelled AA upon LPS stimulation of the cells strongly indicated the previous uptake and incorporation of AA into more complex lipids, from where AA was apparently released by the action of LPS-induced phospholipase A2 [28]. In order to test potential biological effects of eicosanoids on U937 cells, a decision was made in favor of commercially available eicosanoids, 5-HETE and 15-HETE, which were found to show remarkable concentration differences among the FCS batches. Hydroxyeicosatetraenoic acids (HETEs) are formed with AA by the action of lipoxygenases ALOX5 and ALOX15, expressed typically by epithelial cells as well as phagocytes such as neutrophils and macrophages [29,30]. Beside their effects on cell proliferation and differentiation, they are known activators of PPARγ [31]. Actually, peroxisome proliferator-activated receptors are known to induce the uptake and metabolism of fatty acids and to strongly modulate immune functions [32]. As fatty acid metabolism takes place in peroxisomes [33], the 5-HETE/15-HETE induced up-regulation of PEX16 (Figure 4), a peroxisome biogenesis protein indicative for peroxisome proliferation [18], indicated that this treatment caused an increased demand for these organelles. The concomitant down-regulation of PKM (Figure 4), a key enzyme for glycolysis [34], may suggest that HETE-treatment of U937 cells induced a metabolic shift increasing beta oxidation and attenuating glycolysis. This interpretation was independently supported by the concentration-dependent HETE-induced formation of peroxisomes (Figure 4) observed by immunofluorescence staining using a PMP70 antibody [35]. 

## 4. Conclusions and Outlook

The present data demonstrate that batch-dependent differences of eicosanoids contained in FCS may have a profound effect on cellular functions as observed with the U937 in vitro cell culture model for differentiation and inflammatory stimulation. Eicosanoids affect many relevant cellular events far beyond that, suggesting that they may represent the main contributors for reproducibility issues in cell culture. The establishment of a strict quality control regime controlling eicosanoid content in FCS may alleviate this challenging problem.

## 5. Materials and Methods

### 5.1. Cell Culture 

U937 cell line was cultured in RPMI medium (1X with L-Glutamine; Gibco, Thermo Fischer Scientific, Vienna, Austria) supplemented with 1% Penicillin/Streptomycin (Sigma-Aldrich, Austria) and 10% Fetal Calf Serum (FCS, Sigma-Aldrich, Vienna, Austria) in T25 polystyrene cell culture flasks for suspension cells (Sarstedt, Wiener Neudorf, Austria) at 37 °C and 5% CO_2_. Cells were counted with a MOXI Z Mini Automated Cell Counter (ORFLO Technologies, Ketchum, ID, USA) using Moxi Z Type M Cassettes (ORFLO Technologies, Ketchum, ID, USA) and the number of seeded cells for the experiments calculated based of these results. For all experiments the cells were used in passages 22–26. 

### 5.2. Differentiation with Phorbol 12-Myristate 13-Acetate (PMA) and Inflammatory Activation with Lipopolysaccharides (LPS)

All experiments were carried out in triplicates of LPS activation and control. For the proteomics and eicosadomics measurements 2 × 10^6^ cells were seeded in T25 polystyrene cell culture flasks with cell growth surface for adherent cells (Sarstedt, Wiener Neudorf, Austria) with 5 mL fully supplemented media and 100 ng/mL PMA (Phorbol 12-myristate 13-acetate ≥ 99%, Sigma-Aldrich, Vienna, Austria) to induce differentiation. After 48 h incubation the medium was withdrawn and used for eicosanoid measurements. Three ml of new fully supplemented media was added either with 1 µg/mL LPS (Lipopolysaccharides from Escherichia coli 055:B5, γ-irradiated, BioXtra, Sigma-Aldrich, Vienna, Austria) or 1 µL PBS per 1 mL medium as control. After 24 h activation the medium was withdrawn again and used for eicosanoid measurements. The cells were gently washed twice with 5 mL phosphate buffered saline (PBS) and 3 mL new medium without FCS was added and incubated. After 4 h the supernatant was withdrawn and used for proteomics measurements. The cells were used for a subcellular fractionation as described before and cytoplasm and nuclear fraction were used for proteomics analysis [36].

### 5.3. Test of Different FCS Batches

Throughout the experiments different suppliers and batches of FCS were used. Additional details concerning the FCS batches are listed in Table 1. FCS batches A-C were heat inactivated at 56 °C for 30 min, batch D was already bought heat inactivated. Also different concentrations of HETEs and labelled arachidonic acid were supplemented and the respective controls treated with the same amount of LC-MS grade methanol (5 µL/3 mL medium). The experimental workflow of PMA differentiation and LPS activation was done for every condition similarly, only exchanging the FCS batch, supplier or eicosanoid. Additionally, for every condition 3 aliquots (3 × 3 mL) of the fully supplemented media were used for eicosanoid measurements to determine the default levels of eicosanoids present.

### 5.4. C13 Labelled Arachidonic Acid

For the investigation of the uptake and release of PUFAs an experiment was carried out with the supplementation of 1,2,3,4,5-^13^C arachidonic acid (C13 AA, Cayman chemicals, Ann Arbor, Michigan, USA). Whenever C13 AA was added, a control experiment was supplemented with the same concentration of unlabeled arachidonic acid. For the first experiment the C13 AA was added at a concentration of 1.6 µM (used for all AA supplementations) to the fully supplemented medium during the 48 h PMA differentiation step. This concentration is around double that of the endogenous arachidonic acid, thus the supplementation tripled the concentration of biologically active arachidonic acid. Afterwards, the now adherent cells were washed three times with PBS and medium without supplemented AA was added together with or without LPS for 24 h. For the second experiment the cells were differentiated with PMA in standard medium, washed three times with PBS and medium supplemented with C13 AA was added together with and without LPS for 24 h. The eicosanoids were collected and measured in the supplemented medium without incubation (t0), after 48 h PMA differentiation and after 24 h LPS activation. The experimental setup is illustrated in Figure 3. 

### 5.5. Proteomics of Supernatant (SN), Cytoplasm (CYT), and Nuclear Extract (NE)

For the proteomics sample preparation, the s-trap system (Protifi, Huntington, NY) was employed following the manufacturers protocol with slight modifications. The precipitated proteins were dissolved in lysis buffer (8 M Urea, 0.05 M triethylammonium bicarbonate (TEAB) and 5% sodium dodecyl sulfate SDS) and diluted to obtain a protein concentration of about 1 µg/µL. Twenty µg of protein was used for each digestion. First, the sample was reduced with 20 µL dithiothreitol DTT (Sigma-Aldrich) at a final concentration of 32 mM for 10 min at 95 °C. Afterwards, 5 µL iodoacetamide IAA (Sigma-Aldrich) was added to a final concentration of 54 mM and incubated for 30 min at 30 °C in the dark. After adding 4.5 µL 12% ortho-phosphoric acid (Sigma-Aldrich) and 297 µl S- Trap buffer (90% Methanol (*v*/*v*) in H_2_O and 0.1 M TEAB) the sample was loaded onto the S-Trap Filter. The S-trap filters were centrifuged at 4000 xg for 1 min to pass through all the sample and trap the proteins onto the resin and afterwards washed four times with 150 µL S-Trap buffer. Twenty µg aliquots of Trypsin/Lys-C (MS grade; Promega Corporation, Madison, WI, USA) were dissolved in 400 µL 50 mM TEAB and 20 µg of this solution was added directly onto the resin of the filter (corresponding to 1 µg Trypsin/Lys-C per sample) and incubated for 1 h at 47 °C. After finishing the digestion, the peptides were eluted with 40 µL of 50 mM TEAB followed by 40 µL of 0.2% formic acid (FA) in H_2_O and 35 µL of 50% (*v*/*v*) acetonitrile (ACN) with 0.2% FA in H_2_O. The peptides were dried for about 2 h with vacuum centrifugation and stored at −20 °C until LC-MS/MS measurement. 

### 5.6. HPLC-MS/MS for Proteomics

For the HPLC-MS/MS analysis the peptides were resolved in 5 µL 30% formic acid and diluted with 40 µL of mobile phase A (97.9% H_2_O, 2% acetonitrile, 0.1% formic acid). One µL for the supernatant samples and 5 µL of cytoplasmic and nuclear samples were injected into the Dionex UltiMate 3000 RSLCnano liquid chromatography (LC) system coupled to the QExactive Orbitrap MS (all Thermo Fisher Scientific, Austria). Peptides were trapped on a C18 2 cm × 100 μm precolumn and LC separation was performed on a 50 cm × 75 μm Pepmap100 analytical column (both Thermo Fisher Scientific, Austria). Separation was achieved applying a 43 min gradient from 7% to 40% mobile phase B (79.9% acetonitrile, 20% H_2_O, 0.1% formic acid) for supernatant samples and 95 min gradients from 8% to 40% mobile phase B for cytoplasmic and nuclear samples, both at a flow rate of 300 nL/min, resulting in a total run time of 85 min and 135 min, respectively. Mass spectrometric settings were the same for all fractions. The resolution on the MS1 level was set to 70,000 (at *m/z* = 200) with a scan range from 400 to 1400 *m/z*. The top eight abundant peptide ions were chosen for fragmentation at 30% normalized collision energy and resulting fragments analyzed in the Orbitrap at a resolution of 17,500 (at *m/z* = 200).

### 5.7. Proteomics Data Analysis

Raw data were subjected to the freely available software MaxQuant (version 1.6.0.1) [37] utilizing the Andromeda search engine, which returns label free quantification (LFQ) values for each identified protein as subsequently used for further data evaluation. For the MaxQuant search, a minimum of two peptide identifications, at least one of them being a unique peptide, was required for valid protein identification. Digestion mode was set to “Specific” choosing Trypsin/P. The peptide mass tolerance was set to 50 ppm for the first search and to 25 ppm for the main search. The false discovery rate (FDR) was set to 0.01 both on peptide and protein level. The database applied for the search was the human Uniprot database (version 03/2018, reviewed entries only) with 20,316 protein entries. Further settings for the search included carbamidomethylation as fixed modification and oxidation of methionine and acetylation of the protein C terminus as variable modifications. Each peptide was allowed to have a maximum of two missed cleavages and two modifications, “Match between runs” was enabled and the alignment window set to 10 min, with the match time window of 1 min. Statistical evaluation was performed with Perseus software (version 1.6.0.2) [38] using LFQ intensities of the MaxQuant result file. After filtering potential contaminants, the LFQ values were Log(2)-transformed. Technical duplicates were averaged. Only proteins detected in three of three biological replicates in either control and/or treatment groups were considered for data evaluation. Permutation-based FDR was set to 0.05 for *t*-tests and provided significant protein expression changes corrected for multi-parameters (S0 = 0.1). The mass spectrometry proteomics data were deposited in the ProteomeXchange Consortium (http://proteomecentral.proteomexchange.org) via the PRIDE partner repository [39] with the dataset identifier PXD020617 and 10.6019/PXD020617.

### 5.8. Eicosanoid Sample Preparation

Cell supernatants were spiked with 5 µL of internal standards (Appendix A) and centrifuged at 726 g for 5 min to remove cells and debris. Three ml of the supernatant was mixed with 12 mL of ice cold ethanol and left at −20 °C overnight to precipitate the contained proteins. The samples were centrifuged for 30 min with 4536 xg at 4 °C and the supernatant transferred into a new 15 mL Falcon tube. Ethanol was evaporated via vacuum centrifugation at 37 °C until the original sample volume was restored. Samples were loaded on conditioned 30 mg/mL StrataX solid phase extraction (SPE) columns (Phenomenex, Torrance, CA, USA). Columns were washed with 2 mL MS grade water and eicosanoids were eluted with 500 µL methanol (MeOH abs.; VWR International, Vienna, Austria) containing 2% formic acid (FA; Sigma-Aldrich). MeOH was evaporated using N_2_ stream at room temperature and reconstituted in 150 µL reconstitution buffer (H2O/ACN/MeOH + 0.2% FA—65:31,5:3,5), containing a second set of internal eicosanoid standards at a concentration of 10–100 nM (Appendix A).

### 5.9. UHPLC-MS/MS for Eicosanoid Measurements

Analytes were separated using a Thermo Scientific Vanquish (UHPLC) system and a Kinetex^®^ C18-column (2.6 μm C18 100 Å, LC Column 150 × 2.1 mm; Phenomenex^®^). Applying a 20 min gradient flow method, starting at 35% B steadily increasing to 90% B (1–10 min), going up to 99% B in 0.25 min. Flow rate was kept at 200 µL/min, 20 µL injection volume and column oven temperature was set to 40 °C. Eluent A contains H_2_O + 0.2% FA and eluent B ACN:MeOH (90:10) + 0.2% FA.

Mass Spectrometric analysis was performed with a Q Exactive HF Quadrupole-Orbitrap mass spectrometer (Thermo Fisher Scientific, Austria), equipped with a HESI source for negative ionization. Mass spectra were recorded operating from 250 to 700 *m*/*z* at a resolution of 60,000 @ 200 *m*/*z* on MS1 level. The two most abundant precursor ions were selected for fragmentation (HCD 24 normalized collision energy), preferentially molecules from an inclusion list which contained 32 *m/z* values specific for eicosanoids (Appendix A). MS2 was operated at a resolution of 15,000 @ 200 *m*/*z*. For negative ionization, a spray voltage of 2.2 kV and a capillary temperature of 253 °C were applied, with the sheath gas set to 46 and the auxiliary gas to 10 arbitrary units.

Generated raw files were analyzed manually using Thermo Xcalibur 4.1.31.9 (Qual browser), comparing reference spectra from the Lipid Maps depository library from July 2018 [40]. For peak integration and quantitative data analysis the software TraceFinder^TM^ (version 4.1-Thermo Scientific, Austria) was used. For the quantification of arachidonic acid (Figure 3), a calibration curve was generated (Appendix A).

### 5.10. Immunofluorescence

For fluorescence microscopy 8 × 10^4^ cells in 400 µL were seeded in a µ-Slide 8 well (Ibitreat coating, ibidi GmbH Martinsried, Germany). Differentiation of the cells was induced with 100 ng/mL PMA for 48 h. Afterwards the cell supernatant was exchanged with fully supplemented medium without PMA for an additional 24 h. Sample preparation was performed as previously described with minor modifications [41]. Cells were fixed with pre-warmed formaldehyde (3.7%) for 15 min and permeabilized with Triton-X 100 (0.2%) for 10 min. Blocking was performed with Donkey serum (2% in PBS-A) for 1 h, room temperature (RT). Primary antibodies were incubated 2h at RT at dilution 1:500. After washing, specie-specific fluorescent-labelled secondary antibodies were added and slides incubated in a dark humidified chamber for 1.5 h. For our study, Anti PMP70 Antibody (Rabbit polyclonal, PA1-650) and Anti TOM20 (F-10, Mouse Monoclonal Sc-17764), Alexa Fluor 488 Donkey Anti Mouse (A21202_LOT2090565) and Alexa Fluor 568 Donkey Anti-Rabbit (A10042_LOT2136776) were used. The slides were washed and post-fixed with 3.7% formaldehyde (10 min, RT); at the end of the post-fixation, 100 mM glycine was used to mask reactive sites and slides were mounted and sealed with Roti-Mount FluoCare (Roth, Graz, Austria) with DAPI. SIM Images were acquired with a Confocal LSM Zeiss 710 equipped with ELYRA PS. 1 system. Structured Illumination Microscopy (SIM) images were obtained with (Plan Apochromat 63X/1.4 oil objective) grid 5 rotation. For the quantification of fluorescence intensities (Figure 4B), 30 optical fields/region of interest (ROI) were quantified for every experimental condition from at least 3 independent experiments.

### 5.11. Differentiation Status by Flow Cytometry

In order to confirm the differentiation status obtained via PMA treatment the cells were tested for the differentiation marker CD11b (ITGAM) using FACS analysis. Therefore, U937 cells were treated with 100 ng/mL PMA for 48 h using 2 × 10^5^ cells per well in 6-well plates. After the incubation time, the cells were washed three times with PBS and put on ice. The differentiation status was assessed by labelling with an anti-CD11b antibody (APC clone D12, BD Bioscience) and subsequent evaluation of the CD11b+ population. Three biological replicates were analyzed per condition on an FACS Canto II cytometer (BD Bioscience).

## Figures and Tables

**Figure 1 biomolecules-11-00113-f001:**
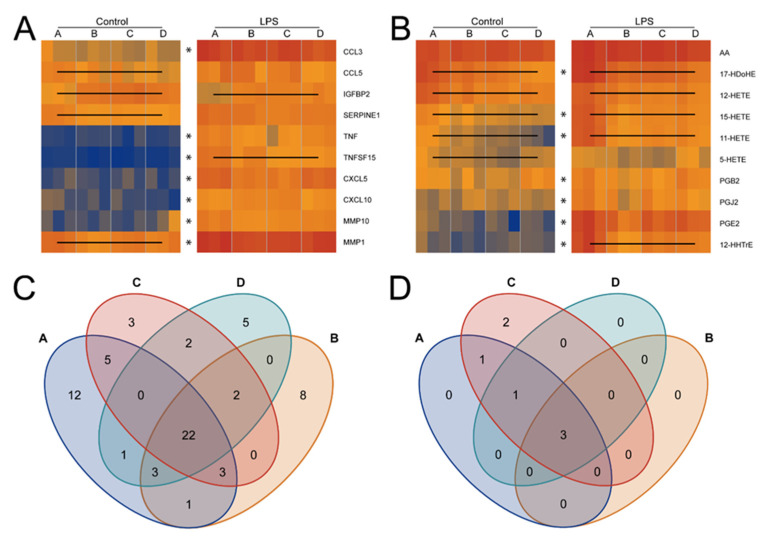
Heatmaps of selected proteins (**A**) and eicosanoids (**B**) determined in secretomes of control and LPS-treated U937 cells cultured with the indicated batch of FCS, A, B, C or D. Lines within a heatmap indicate a significant difference of the given molecule within at least two batches. Asterisks (*) indicate that LPS-treatment induced a significant increase. Venn diagrams of significantly up- and downregulated (**C**) proteins (S0 = 2, FDR = 0.01) and (**D**) eicosanoids comparing LPS activation with control samples for all four FCS batches (**A**–**D**).

**Figure 2 biomolecules-11-00113-f002:**
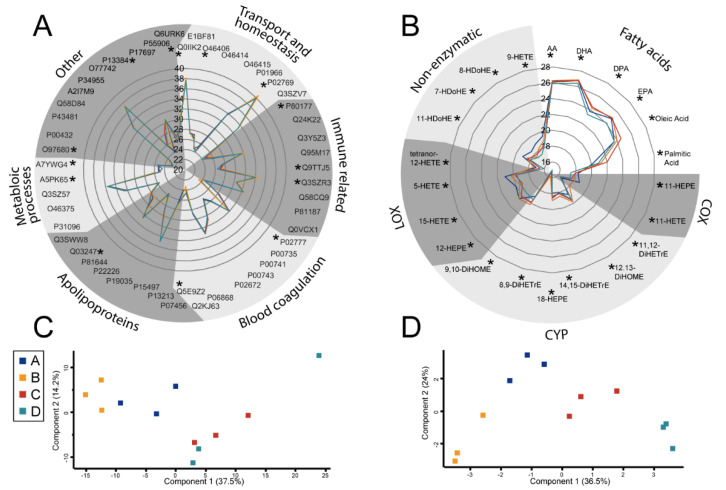
Radar plot for selected proteins (**A**) and fatty acids (**B**) identified in 4 different FCS batches without incubation with cells (baseline levels). Principal component analysis of protein (**C**) and eicosanoid (**D**) measurements of the same FCS batches, as indicated by different colors, demonstrates superior clustering in the case of eicosanoids. Asterisks mark significantly regulated molecules.

**Figure 3 biomolecules-11-00113-f003:**
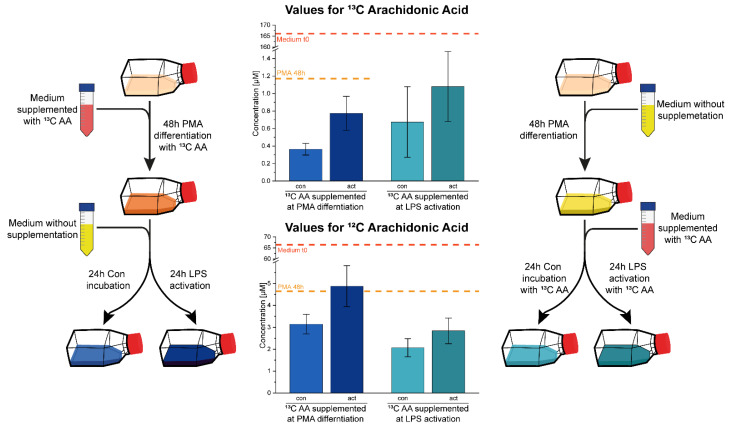
Experimental setup and results from AA spiking experiments. Medium was supplemented with 13C AA either before differentiation (left hand side) or before LPS treatment (right hand side). AA determination of cell supernatants by LC-MS/MS revealed AUC values as indicated. Medium levels at the beginning of cell culture are indicated by lines. Error bars are derived from three independent experiments. Con, untreated cells; act, LPS-treated cells. Note that AA concentration values strongly decrease upon cell cultivation but increase again upon LPS treatment.

**Figure 4 biomolecules-11-00113-f004:**
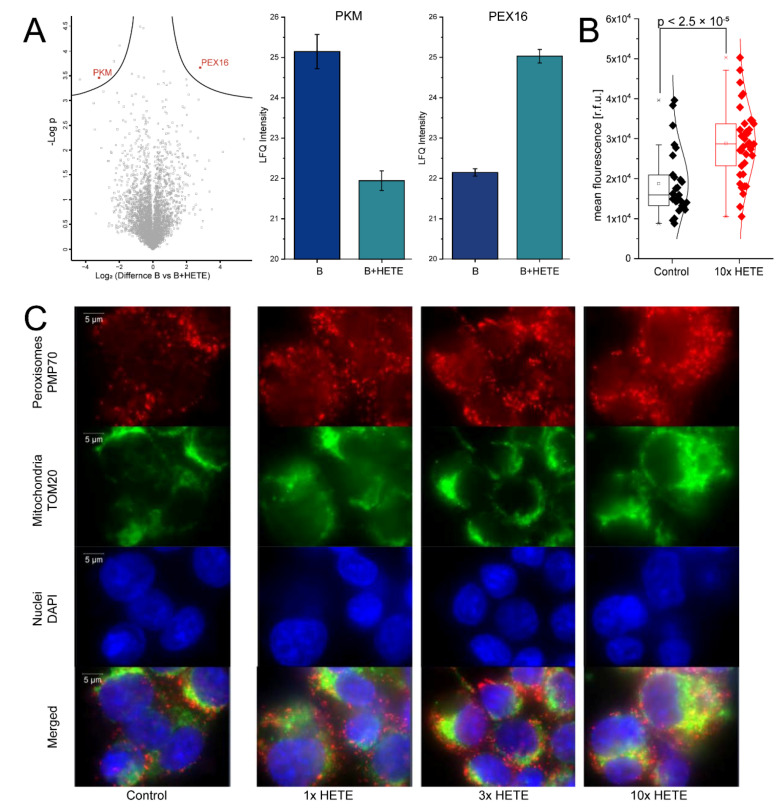
(**A**) Eicosanoid levels of 5-HETE and 15-HETE for FCS-B (before spiking), FCS-A and FCS-B after spiking with 5-HETE and 15-HETE. (**B**) Volcano plot for cytoplasmic proteins obtained from U937 cells after PMA differentiation when cultured in either FCS-B or FCS-B supplemented HETEs. Bar plots exemplify the significantly regulated proteins PKM and PEX16. (**C**) Immunofluorescence detection of peroxisomes (red, PMP70 antibody), mitochondria (green, TOM20 antibody) and nuclei (blue, DAPI) shown for control and increasing concentrations of supplemented HETEs (addition of 1, 3, or 10 times of the spiked HETE mix).

**Table 1 biomolecules-11-00113-t001:** Tested FCS batches stating Vender, Lot number, expiration date as well as letter used in this work.

Nomenclature	Vendor	Lot Number	Expiration Date	Origin	Processed
A	Sigma	BCBT4187	07.2021		-
B	Gibco	42Q5650K	06.2020	Brazil	-
C	Gibco	42G8378K	11.2022	Brazil	-
D	Gibco	08Q8082K	02.2023	Brazil	Heat inactivated

## Data Availability

The mass spectrometry proteomics data were deposited in the ProteomeXchange Consortium (http://proteomecentral.proteomexchange.org) via the PRIDE partner repository [39] with the dataset identifier PXD020617 and 10.6019/PXD020617.

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
