# Peer review of "Eicosanoid Content in Fetal Calf Serum Accounts for Reproducibility Challenges in Cell Culture"

_biomolecules, 2021, doi:10.3390/biom11010113_

Round 1

Reviewer 1 Report

In the manuscript titled "Eicosanoid content in fetal calf serum accounts for reproducibility challenges in cell culture" by Niederstaetter et al., authors report that the differences in the concentration of eicosanoids between batches of fetal calf sera leads to substantial differences in cell culture outcomes.  While the manuscript is well written and presented, some method details are important to be provided for readers and reviewers to appreciate the results presented here.

(1) Authors should provide detailed description of methods employed to quantitate and ascertain the statistical significance of the protein and fatty acid concentrations.

(2) Heatmaps presented in the figure 1A & 1B should indicate individual replicates for each of the batches and corresponding data table should be provided as supplementary material.

(3) With respect to Figure 1C & 1D, please mention about how was the list of Up and Down proteins and eicosanoids generated.

(4) At line 271 authors mentioned that Figure 2D(Eicosanoids) demonstrated more variation than Figure 2B (Proteins). However, PC1 of Figure 2B (37.5%) shows more variance than PC1 of Figure 2D(36.5%). PC1 is considered the axis of maximum variance. Authors should clarify this statement.

Author Response

We would like to thank the reviewer for careful evaluation and helpul comments. We have now provided all requested information as outlined below in more detail.

In the manuscript titled "Eicosanoid content in fetal calf serum accounts for reproducibility challenges in cell culture" by Niederstaetter et al., authors report that the differences in the concentration of eicosanoids between batches of fetal calf sera leads to substantial differences in cell culture outcomes.  While the manuscript is well written and presented, some method details are important to be provided for readers and reviewers to appreciate the results presented here.

We would like to thank the reviewer for the positive comments.

(1) Authors should provide detailed description of methods employed to quantitate and ascertain the statistical significance of the protein and fatty acid concentrations.

            We thank the reviewer for pointing out this important lack of experimental detail, which was now corrected accordingly. For the assessment of abundance values and statistical significance, we have applied label free quantification strategies and false discovery calculation methods as indicated in Materials and Methods, section “Proteomics Data Analysis”. In case of eicosanoids, the software package TraceFinderTM was used for quantification as indicated. We have also specified the employed quantification procedures in more detail (Materials and Methods, sections “UHPLC-MS/MS for eicosanoid measurements” and “Immunofluorescence”).

(2) Heatmaps presented in the figure 1A & 1B should indicate individual replicates for each of the batches and corresponding data table should be provided as supplementary material.

            The Figure has been updated accordingly, now showing individual replicates. Please note that Supplementary Tables already include all individual measurement results, while the experimental evidence for each data point are provided by the data made public via ProteomeXchange.

(3) With respect to Figure 1C & 1D, please mention about how was the list of Up and Down proteins and eicosanoids generated.

            Please find the requested information in our answer to the first question raised. In general, we have applied label-free quantification strategies evaluated with MaxQuant and Perseus for proteomics data and TraceFinder for eicosanoids.

(4) At line 271 authors mentioned that Figure 2D(Eicosanoids) demonstrated more variation than Figure 2B (Proteins). However, PC1 of Figure 2B (37.5%) shows more variance than PC1 of Figure 2D(36.5%). PC1 is considered the axis of maximum variance. Authors should clarify this statement.

            We again thank the reviewer for this careful consideration. However, if you looked at the distances between the centers of the groups formed by each batch compared to distances between replicates (within versus between), it gets obvious that in case of eicosanoids the distances between groups are clearly larger than distances between replicates, while in case of proteins these distances are rather similar. Thus, the variance between groups is larger in case of eicosanoids as compared to proteins. This interpretation is supported by the observation of more similar abundance levels determined in case of proteins when compared to eicosanoids (colored lines in Figure 2).

Reviewer 2 Report

The manuscript from Niederstaetter et al. deals with the effects caused in cultured cells by media supplementation with different batches of fetal calf serum (FCS). These cellular effects were hereby mainly evaluated by proteomic and metabolomic approaches, as well as, proteomic and metabolomic techniques were also used to analyze the composition of the different FCS batches. Among the various classes of FCS constituents, the authors particularly focused their attention onto proteins and eicosanoids as such molecules are considered by the authors the more likely candidates for cellular effects caused by FCS batch-dependent variability. More in particular, the authors focus onto the role of eicosanoids as inflammatory mediators. In fact, the authors used U937 macrophages as a cellular model where differentiation and inflammatory activation were properly induced. The authors ingeniously used LC-MS/MS techniques to highlight differences in the cell proteomes and secretomes as well as in the protein and eicosanoid content of FCS batches. The authors also used media over-supplementation with arachidonic acid (AA, an eicosanoid precursor molecule) to study the metabolism of such molecule also in response to the inflammatory activation of U937 cells. Moreover, the authors also performed functional analyses after media supplementation with two hydroxyeicosatetraenoic acids (HETE), namely 5-HETE and 15-HETE, respectively. In fact, HETEs are formed starting from AA and the levels of 5-HETE and 15-HETE were found showing significant differences among the various FCS batches. These latter analyses showed the link between the HETE concentrations and the activation of peroxisomal pathways which was also confirmed by confocal microscopy. More in general, the results suggested that HETEs are able to induce into U937 cells a metabolic change which enhances beta oxidation while decreases glycolysis.

Globally, this work shows that different batch-dependent contents of eicosanoids in FCS may be responsible of relevant differential effect on cellular functions. Moreover, the results of this work suggest that in future the reproducibility of most experiments in cell culture would take advantage of more attention about cell media supplementation. For this reason, some of the hints given by this work may be important for most of the research in the biomedical field.

The present work is well planned and well performed.

The manuscript is well structured and well written.

I strongly recommend this paper for publication into this journal.

However, I have some minor comments:

  • At line 59, the sentence “….appeared less likely as it should be limited….” is not very clear. It is worth to be rephrased.
  • The panel C of the Figure 4 would be clearer if the images of all the other single fluorescence channels (TOM20 and DAPI signals) were shown (and not only the signal of PMP70 and the merge). Moreover, about the panel B of the same figures the authors should specify (in the caption or even in the Materials and Methods section) the number of: replicates, microscope fields per replicate and focal planes per field.
  • The format of some powers and exponents should be corrected (e.g. line 217 “…2 x 105 cells…”).
  • More attention should be given to the abbreviations. Some are used without specify the term (e.g. RT for “room temperature”). Maybe some abbreviations are not worth to be included in the appropriate section “Abbreviations”. However, the abbreviations which are not included in such list should be specified on the first time that the relative term appears in the manuscript. Example: at line 175, “room temperature” should become: “room temperature (RT)”. This is not the only case. There are other abbreviations which should be correctly specified.

Author Response

We would like to thank this reviewer for the positive and helpful comments. We have now included all requested information as outlined below in more detail.

  • At line 59, the sentence “….appeared less likely as it should be limited….” is not very clear. It is worth to be rephrased.

We have rephrased the sentence to make it clearer.

  • The panel C of the Figure 4 would be clearer if the images of all the other single fluorescence channels (TOM20 and DAPI signals) were shown (and not only the signal of PMP70 and the merge). Moreover, about the panel B of the same figures the authors should specify (in the caption or even in the Materials and Methods section) the number of: replicates, microscope fields per replicate and focal planes per field.

We have now provided the independent fluorescence channels and detailed the number of microscopic fields for data evaluation

  • The format of some powers and exponents should be corrected (e.g. line 217 “…2 x 105 cells…”).

These errors were corrected

  • More attention should be given to the abbreviations. Some are used without specify the term (e.g. RT for “room temperature”). Maybe some abbreviations are not worth to be included in the appropriate section “Abbreviations”. However, the abbreviations which are not included in such list should be specified on the first time that the relative term appears in the manuscript. Example: at line 175, “room temperature” should become: “room temperature (RT)”. This is not the only case. There are other abbreviations which should be correctly specified.

We apologize for this sloppiness and have now specified all abbreviations

Round 2

Reviewer 1 Report

My concerns raised in the previous version of the manuscript have been addressed appropriately.